# Exosome-Laden Hydrogels as Promising Carriers for Oral and Bone Tissue Engineering: Insight into Cell-Free Drug Delivery

**DOI:** 10.3390/ijms252011092

**Published:** 2024-10-15

**Authors:** Cassandra Villani, Prasathkumar Murugan, Anne George

**Affiliations:** Brodie Tooth Development Genetics & Regenerative Medicine Research Laboratory, Department of Oral Biology, University of Illinois at Chicago, Chicago, IL 60612, USA; cvilla61@uic.edu (C.V.); pmurugan@uic.edu (P.M.)

**Keywords:** hydrogels, biomaterials, exosomes, oral tissue engineering, bone tissue engineering, mineralized tissues, regenerative medicine

## Abstract

Mineralization is a key biological process that is required for the development and repair of tissues such as teeth, bone and cartilage. Exosomes (Exo) are a subset of extracellular vesicles (~50–150 nm) that are secreted by cells and contain genetic material, proteins, lipids, nucleic acids, and other biological substances that have been extensively researched for bone and oral tissue regeneration. However, Exo-free biomaterials or exosome treatments exhibit poor bioavailability and lack controlled release mechanisms at the target site during tissue regeneration. By encapsulating the Exos into biomaterials like hydrogels, these disadvantages can be mitigated. Several tissue engineering approaches, such as those for wound healing processes in diabetes mellitus, treatment of osteoarthritis (OA) and cartilage degeneration, repair of intervertebral disc degeneration, and cardiovascular diseases, etc., have been exploited to deliver exosomes containing a variety of therapeutic and diagnostic cargos to target tissues. Despite the significant efficacy of Exo-laden hydrogels, their use in mineralized tissues, such as oral and bone tissue, is very sparse. This review aims to explore and summarize the literature related to the therapeutic potential of hydrogel-encapsulated exosomes for bone and oral tissue engineering and provides insight and practical procedures for the development of future clinical techniques.

## 1. Introduction

Tissue engineering is a branch of biomedical science and multidisciplinary concept often associated with regenerative medicine and has a distinct focus on aspects related to the development of procedures for specific tissue regeneration, maintenance, and/or improved function [1,2]. In addition to correcting congenital defects, regenerative medicine strives to repair or replace tissues at the local and organ level that have been injured or damaged by acute or chronic factors. The potential use of regenerative medicine techniques to treat both chronic and acute injuries, as well as diseases that affect a wide range of organ systems, including traumatic injuries, dermal wounds, cardiovascular diseases, cancer and various procedural complications, has become a popular research field [3]. A variety of techniques, including cell therapy, the administration of therapeutic agents, natural and synthetic biomaterials, polymeric matrix, exosomes, liposomes, different types of nanoparticles and many others that target tissue, aid tissue healing and the repair process [4,5]. Tissue regeneration is accomplished by the use of extracellular matrix (ECM), cells, and a variety of signaling molecules as the core components either individually or combined [3,5].

Mineralization is a key biological process that is required for the development and repair of tissues, such as teeth, bone, and cartilage, which comprises different minerals, such as calcium oxalates, calcium carbonates, and calcium phosphates [6]. Defects of hard and soft craniofacial tissues significantly affect oral functions and cosmetic appearance, in addition to creating treatment-related challenges [7]. Oral and maxillofacial deficiencies can result from a variety of injuries, diseases, and disorders, such as infections, bleeding, cancers, periodontitis, tooth loss, alveolar deterioration, and oral dysfunction [8]. Over the last few decades, transplantation of autologous tissue has been considered the ideal procedure for restoring oral and maxillofacial function. Disadvantages, such as unknown long-term results, morbidity of the donor site, and inadequate tissue quantity, have limited the clinical applications [7]. Bone defects involving the breakdown and/or loss of structural integrity are often caused by trauma from sports or motor vehicle and workplace accidents, as well as bone disorders, such as osteoporosis, osteosarcoma, femoral head necrosis, infections, and aging [9,10]. Therefore, bone deficiencies frequently result in significant disabilities, which severely reduce patients’ quality of life and productivity. Currently, allogeneic and autologous bone grafts are widely used for the treatment of bone defects, among which bone autografts have been recognized as the gold standard. However, the use of autologous bone grafts frequently causes complications, such as localized hematomas and postoperative pain at the donor site [9]. Hence, craniofacial bone defects have emerged as serious illnesses that require alternative effective treatments.

Biomaterials are nondrug substances that can interact with tissues and body fluids for a prolonged period of time and help restore the structure and functions of damaged or impaired tissues or organs. Recently, many biomaterials and composites, including hydrogels, hydrocolloids, foams, scaffolds, fibers fabricated by electrospinning, films, and membranes, were developed with different printing capabilities and properties that increased cell proliferation and migration [11]. Biomaterials have also been used for the diagnosis and treatment of cancer due to their ability to overcome the inherent limitations of conventional methods [12]. Furthermore, modified biomaterials can be developed with different cellular components, such as growth factors, hormones, exosomes, proteins, collagen-like peptides, selection of cell types, etc., to mimic native tissues or organs to improve cellular signaling or extracellular matrix (ECM) formation [11,13]. During natural bone regeneration, the ECM has proven crucial for both signaling and material exchange between the native tissue and regenerated tissue [14,15]. Hydrogels have been shown to be an excellent scaffold for promoting cell development and loading for the sustained release of drugs or bioactive compounds. As water-swelling polymers with a 3D network structure formed by several crosslinking events, hydrogels are also able to create a regulated microenvironment, even in confined spaces. Thus, hydrogel biomaterials have an inherent advantage over other biomaterials in regenerative tissue engineering [14].

Exosomes (Exo) are a subset of extracellular vesicles (~50–150 nm) that are secreted by cells and contain genetic material, proteins, lipids, nucleic acids, and other biological substances that have been extensively researched for bone and oral tissue regeneration [16,17,18,19]. They have been found to play an essential role in many biological and pathological processes. Engineered exosomes can be used for targeted delivery. Exosomes derived from mesenchymal stem cells (MSCs-Exo) are used extensively in regeneration of oral tissues due to their ability to promote repair of both mucosal tissue and alveolar bone as well as regulate immune disorders and inhibit inflammation [20,21]. Exosome binding to the cell membrane is crucial to their activity as it allows them to be endocytosed, thus transferring their biological cargo to cells in their proximity. The main advantage is that they are biocompatible, nonimmunogenic, nontoxic, and unlike the use of stem cells, they are free from tumorigenic potential and ethical issues [22]. As exosomes are nanosized particles, they make excellent carriers to deliver a variety of cargo, including drugs, nucleic acids, proteins, and gene therapy agents, even across the blood–brain barrier. Studies have demonstrated that exosomes from MSCs and other cells may influence angiogenesis, immune responses and the regenerative effects of cell therapy, making cell-free treatments a possibility. These characteristics highlight why exosomes have been a popular and promising concept in regenerative medicine and tissue engineering applications [23]. These characteristics highlight why exosomes have been a popular and promising concept in regenerative medicine and tissue engineering applications [23].

Although exosome therapies are promising, they face several challenges, like inherent short half-life and poor standardization of exosome collection techniques and batch-to-batch variations of the molecular cargo resulting in altered functionality due to delivery kinetics, dosing, reproducibility and pharmacokinetics [24]. Thus, there are several gaps in knowledge that need to be ascertained before their use in clinical applications. Nevertheless, cell-derived exosome treatment is a promising therapeutic strategy in regenerative medicine.

The stability of exosomes is important when considered for use in clinical applications. Proteases and nucleases could degrade the proteins and RNA within the vesicles reducing their stability and functionality [25]. By encapsulating the Exo into biomaterials like hydrogels, these disadvantages can be mitigated. Additionally, certain hydrogels inherently have the potential to regulate cellular characteristics, such as adhesion, proliferation and differentiation. Hydrogel biomaterials can be engineered to functionally improve bioavailability, biocompatibility, and not only sustained but controlled release at the site of injury [26]. Both natural and synthetic hydrogels have been used for embedding exosomes. The properties of hydrogels, including interconnected networks, small pores, adjustable structure, and gradual degradation, support exosome retention while sustaining long-term delivery. This helps increase the availability of Exo at the target site and reduce the complications that can arise from rapid administration. Another advantage of Exo-loaded hydrogels is their ability to be applied in a variety of forms, including injectable gels [27]. Further, Therefore, the synergistic combination of exosomes and hydrogels would greatly enhance the efficiency of exosomes and also promote tissue repair process. The use of hydrogels fabricated with embedded Exo could be capable of regenerating both bone and oral tissues. Hence, this article aims to explore and summarize the literature related to the therapeutic potential of hydrogel-encapsulated exosomes for bone and oral tissue engineering (Figure 1). Ultimately, it would provide insight and practical methodologies for the development of future clinical techniques.

## 2. Oral Tissue Engineering

Teeth and oral tissues are critical structures for certain functions in humans and animals [28]. The degradation of or lost teeth and the surrounding tissues is a common clinical issue in dentistry and can be caused by dental anomalies, dental caries, periodontitis, traumatic injuries, or systemic disease. Restoration of lost teeth, oral tissue, and maxillofacial structures that can perform and withstand normal functions would provide a huge improvement to the quality of life for patients [28,29,30]. However, the complex anatomical structures of dental, oral and craniofacial (DOC) tissues present a multifaceted challenge for regeneration due to the number of vastly different phenotypes and functions of the facial muscles, tendons, periodontal structures, teeth, temporomandibular joints (TMJ), cranial sutures, bone, cartilage, and salivary glands (Figure 2) [31]. Oral tissue diseases often stem from the interruption or disruption of cell–cell or cell–ECM interactions, and characterization of these changes at either the microscopic or macroscopic level can define the oral defect [32]. The concept of dental tissue regeneration has been under investigation since calcium hydroxide was first proposed for dentin regeneration over 70 years ago. However, it was not until 1991 that regeneration of a temporomandibular joint was attempted [28]. Recent advances in oral tissue engineering and regenerative dentistry have been made due to an increased interest in the fields of molecular engineering, genetics, stem cell biology, as well as diseases that affect various oral structures that have allowed for a greater understanding of the molecular processes responsible for maintaining DOC structures [31,33]. Generally, oral tissue engineering can be separated into two types: mineralized tissue engineering (bone and teeth) and soft tissue engineering (skin, mucosa, and salivary glands). These two subgroups are then individually approached using the three main tissue engineering approaches: conductive vehicle (scaffolds), cell transplantation, and bioactive factors [34]. This has resulted in tissue engineering techniques for the restoration of nearly all tissues in the craniofacial region. The potential for clinical and therapeutic applications of tissue engineering in dentistry is illustrated in Figure 3 [28].

In current clinical practice, implants or dentures are often used to replace lost permanent teeth. However, these restorations do not mimic natural dentition, often contributing to altered mastication and discomfort. A persistent goal in dentistry has been to develop tooth regeneration therapy for both individual tissues as well as whole teeth; a tissue-engineering-based approach has gained popularity for its potential to reduce the drawbacks of artificial restoration and improve patient quality of life [30].

Periodontal disease is one of the main causes of tooth loss around the globe, affecting both hard and soft oral tissues. The periodontium comprises closely connected structures, the periodontal ligament (PDL), cementum and alveolar bone that support oral functions and serve as a barrier against pathogens. The PDL is a fibrous sensory network that connects the alveolar bone to the tooth root surface. It also serves as a source of adult stem cells for maintaining homeostasis of periodontal tissues with normal mastication, after injury and tissue destruction from periodontal disease. The use of tissue engineering techniques to generate 3D-printed tissue models that can uphold the structural and sensory function of native periodontium presents an opportunity to preserve natural dentition [35]. For maxillofacial bone regeneration, a variety of bone substitute materials have been studied in vivo and in the clinic for their ability to stimulate bone growth in defects. Tissue-engineered biomaterials are attractive as bone substitutes because they can fill the injury space while enhancing tissue growth. However, they must satisfy a wide array of criteria, including having a suitable surface for cell adhesion, good biocompatibility and maintaining mechanical stability. 3D-printed scaffolds are often used, as they can hold their shape within the defect while regeneration occurs (mimicking the function of the ECM) and can be printed to precisely fill the bony defect. While this would be advantageous for surgical correction of complex maxillofacial deformities, the continued study of the scaffold material and embedded biological components is necessary for the effective application of these techniques in the clinic [36,37]. Cryogenic 3D-printed hydrogel scaffolds for loading exosomes was shown to be promising in inducing angiogenesis [38]. 3D printing with tissue-specific decellularized ECM and human adipose mesenchymal cell derived-exosomes was shown to promote efficient cartilage and subchondral bone regeneration [39]. Enhanced bone tissue regeneration was observed using a 3D-printed polylactic acid/titanium composite scaffold with plasma treatment [40]. Various other 3D printing technologies with advantages and disadvantages for bone tissue engineering scaffold have been summarized by Zhang et al. [10]. The salivary glands and oral mucosa are also dynamic and complex tissues that support oral function. They comprise ECM networks, vasculature, and varied cell populations that generate specific structures for required physiological signaling. Tissue engineering of the salivary gland and oral mucosa with 3D cell culture models (organoids and spheroids), tissue-on-a-chip models, and functional decellularized scaffolds are all widely studied [41]. These techniques, in conjunction with those for periodontal tissues and bone, are all working to create cutting-edge procedures capable of reproducing the structures and functions of oral tissues (Figure 3).

### 2.1. Hydrogels for Oral Tissue Engineering

Dental biomaterials have evolved from bioinert structures to integrative bioactive materials that aim to support the restoration of oral tissues to their native structure and function. There are still challenges associated with oral tissue regeneration, despite material advancements, due to the tissue variety and individual complexity within the oral cavity [33]. Across the field of dental research, hydrogels as a biomaterial have great potential for both supporting the reconstruction of tissue structure and function as well as treating oral diseases (Figure 4) [42]. Hydrogels are amphiphilic polymer networks that maintain their structural integrity despite their high fluid absorption capacity. Their highly customizable microstructures allow for specific mimicry of cellular ECM including potential for incorporating bioactive molecules [42,43]. The injectable use of hydrogels was initially evaluated in the fields of periodontics and endodontics for their potential antimicrobial properties. This is a critical area of study, as mitigating pathogen infiltration after a root canal or periodontal treatment can determine treatment effectiveness and longevity. When compared to scaffolds with predetermined configurations, which frequently cannot fit into small irregular areas, injectable formulations that can completely fill voids or gaps are ideal [44]. Smart hydrogels, which can respond to external stimuli, such as changes in pH, light, and temperature during wound healing, have been developed, and this could be used for the application of temperature-sensitive hydrogels for wound healing in the oral cavity (~37 °C) [45]. Magnetic-responsive hydrogels that are prepared by embedding magnetic nanoparticles into hydrogel network are advantageous for biomedical applications. The development of magnetic-responsive hydrogels has shown on-demand controllability and responsiveness, as observed in remote-controlled drug delivery [46]. These properties, including the application of light or a magnetic field to the wound surface, can be used to modulate the release of molecules from the hydrogel without causing additional injury.

Dysregulation of the demineralization–remineralization equilibrium at the tooth surface can result in irreversible loss of enamel as mature enamel lacks cells and cannot repair itself. Since hydrogel-based scaffolds have the ability to provide the support required for tissue regeneration, these biomaterials are being revised to mimic enamel and aid in the repair of lost mineralized tissue. However, the physical, functional, and visual properties of hydroxyapatite (HA) crystals present in mature enamel differ from synthetic HA nanorods. Thus far, only extreme laboratory conditions are able to fabricate synthetic HA that can mimic its natural counterpart, further emphasizing the need for the development of biomaterials with more favorable properties and fabrication requirements for tooth remineralization [33]. Administering therapeutic agents into tissues during the regeneration of cementum, dentin, and enamel requires an efficient carrier. Drug delivery systems not only need to be biocompatible, but their degradation properties are critical to the controlled and sustained release of the bioactive molecules they contain. Chitosan is an ideal biocompatible delivery system for local lesions since it is able to create a film on the tooth surface due to its positive charge. Remineralization studies using chitosan-based delivery systems have shown the formation of organized apatite crystals that resemble natural enamel, as well as significant dentin remineralization and cementum formation. Recent advances in nanotechnology have allowed for the design of chitosan-based scaffolds that enhance their natural properties and increase temporal regulation of drug release, leading to new tissue engineering approaches that would be highly beneficial for the stimulation of oral tissue regeneration [47].

Hyaluronic acid hydrogels have received great attention in tissue engineering applications as hyaluronic acid is a natural polysaccharide with low immunogenicity that is one of the main constituents in the ECM throughout the human body. These hydrogels can be used individually or in combination with other polymeric materials, bioactive molecules, or drugs to promote wound healing and prevent infection [48,49,50].

Wang et al. developed a photo cross-linked hyaluronic acid and silk fibrin hydrogel loaded with human dental pulp stem cells (hDPSCs) for pulp regeneration. The hydrogel exhibited cellular growth, proliferation, and osteogenic differentiation ability [51]. DPSC-laden collagen hydrogel matrix have also increased long-term cellular survival by simulating the inner root canal, interacting with the surrounding dentin, and increasing mineralization [52]. Collagen-based hydrogels also exhibit outstanding biological characteristics; they have been shown to promote oral fibroblast growth and proliferation within the hydrogel, as well as adhesion and differentiation of epithelial cells on the substrate surface [53]. Regenerative dentistry has demonstrated the therapeutic efficacy of alginate hydrogels in oral tissue engineering. The alginate hydrogel provides an ideal modality for the sustained release of encapsulated transforming growth factor-β (TGFβ), which can stimulate dentin, pulp and periodontal regeneration. Acid-treated alginate significantly influences the development of odontoblast-like cells and functions as a dentinal extracellular matrix. Furthermore, an alginate scaffold can serve as a support material for successful prosthetic dental implantation when coupled with nano-bioglass ceramic [54].

### 2.2. Exosomes for Oral Tissue Engineering

Oral and maxillofacial injuries or diseases often need regenerative therapy. While stem cell applications in dental tissue engineering are receiving much attention, there are still obstacles to these applications in clinical settings. Thus, cell-free-based tissue engineering employing exosomes isolated from stem cells has been proposed to be a potential alternative (Figure 5A) [55,56]. Exosomes have been extensively utilized to treat oral and maxillofacial conditions; however, biological exosomes have limited therapeutic efficiency due to their high impurities, poor yield, time-intensive isolation, and lack of targeting. Recent research has used engineering techniques to enable the precise application of exosomes in the treatment of oral and maxillofacial disorders. Engineered stem cell exosomes have also shown positive therapeutic effects in accelerating oral and maxillofacial wound repair by controlling inflammation, improving angiogenesis, fibroblast proliferation, and reducing scar formation [57,58]. Engineered exosomes have also shown potential for the diagnosis of periodontitis. Research has indicated that those with periodontitis have considerably lower amounts of CD9 and CD81 in their salivary exosomes [59].

Small extracellular vesicles generated by hDPSCs (DPSC-Exos) have regenerative abilities in a mildly inflammatory microenvironment and can promote dental pulp regeneration through functional healing rather than scar healing, which could have implications in regenerative endodontics [60]. The DPSC-Exos could also promote the cell migration, proliferation, and osteogenic differentiation of periodontal ligament stem cells (PDLSCs). In addition, DPSC-Exos have been shown to improve the periodontitis microenvironment and help in regulating the macrophage phenotypes. The study by Qiao et al. demonstrated that DPSC-Exos exert anti-inflammatory effects via the IL-6/JAK2/STAT3 signaling cascade, reducing expression after induction via LPS. This was further evaluated with an in vivo repair model using rats with periodontitis. The disease model was generated via a ligature treated with LPS from Porphyromonas gingivalis. Compared to the control periodontitis treatment with DPSC-Exos resulted in an increased periodontal epithelium thickness and alveolar bone height [61].

### 2.3. Exosome Laden Hydrogels for Oral Tissue Engineering

In order to deliver exosomes to the injured or damaged tissues, many strategies have been explored. The ability to administer a high dose of exosomes easily is essential to the efficacy of exosome therapy. Therefore, to attain a high therapeutic efficacy of exosomes, the mode of exosome delivery needs to be optimized and improved. Throughout the past decade, hydrogel-based exosome delivery systems have been the focus of research for their potential use in regenerative medicine [62]. Further, exosomes can be used to promote oral and maxillofacial wound healing by altering the wound inflammatory microenvironment, promoting re-epithelialization, and enhancing angiogenesis [45]. In oral regenerative medicine, the Exo-embedded hydroxypropyl chitin (HPCH)/chitin whisker (CW) thermosensitive injectable hydrogel was able to improve hDPSC angiogenesis and odontogenesis potential. The implantation of a HPCH/CW tooth root model in an in vivo animal experiment demonstrated the development of new tissues that mimicked dental pulp [63]. In another study, DPSC-Exos were encapsulated in a peptide hydrogel and administered to nude mice. After eight weeks, the mice developed pulp-like tissue consisting of mineralized tissue and collagen fibers that were correctly aligned with a root canal wall that contained numerous interconnected blood vessels and nerves. To avoid limiting the therapeutic effects of exosomes, it has been observed that hydrogels could protect free exosomes in vivo from rapid degradation by the immune system, while also ensuring targeted delivery This would maintain the concentration of exosomes reaching the target tissues and extend their residence time [64].

In order to better adhere to modified collagen within the hydrogel and facilitate the wrapping and delayed release of exosomes, the exosomes have a binding site for type I collagen and cell adhesion proteins on the surfaces of their cell membranes [64]. For instance, Zhang et al. created a collagen hydrogel encapsulated with epithelial root sheath cell-derived exosomes that improved angiogenesis and odontogenic differentiation of dental papilla cells. In an in vivo tooth root slice model, the degradation of the hydrogel and subsequent release of exosomes triggered the regeneration of both soft (neurons and blood vessels) and hard (reparative dentin-like tissue) tissues [65]. Another study found that wrapping DPSC-Exos in a fibrin/gelatin hydrogel enabled the slow release of exosomes due to the gradual degradation of the gelatin. This sustained release of exosomes promoted stem cell differentiation and an increase in blood vessel formation until dental pulp-like tissue was formed in the fibrin/gelatin environment [61]. In vitro, exosomes generated from immortalized murine odontoblast cells (MDPCs) and hDPSC enhanced mineralization and increased the expression of odontogenic genes. An amphiphilic synthetic polymeric vehicle from a triblock copolymer made of polyethylene glycol-poly lactic-co-glycolic acid (PEG-PLGA), was able to encapsulate exosomes by polymeric self-assembly and maintain their biological stability throughout release up to 8–12 weeks. After six weeks, the regulated release of odontogenic exosomes produced a reparative dentin bridge in a rat molar pulpotomy model that was superior to glass-ionomer cement alone [66].

While many antibacterial treatments using hydrogels have been proposed and investigated, researchers have noted that the degeneration of periodontal tissue ultimately results from dysregulation of the host immune–inflammatory response mediated by pro-inflammatory macrophages [67]. Bacterial infection is only the first factor in the progression of periodontitis. Treatments for the periodontal system are typically ineffective for a sensitive host with severe periodontitis because pro-inflammatory cytokines, such as TNF-α and IL-1, continue to drive alveolar bone resorption and inflammation of the periodontium [67,68,69,70]. Shen et al. developed a chitosan hydrogel loaded with dental pulp stem cells derived from Exso (DPSC-Exo/CS) for periodontitis (Figure 5B). The DPSC-Exo/CS hydrogels in this study accelerated the repair of the periodontal epithelium and alveolar bone in mice with periodontitis. The hydrogel specifically assisted the shift of macrophages from a pro-inflammatory to an anti-inflammatory phenotype. It was suggested that this mechanism could be connected with miR-1246 in DPSC-Exos. These findings not only highlight the DPSC-Exo/CS therapeutic mechanism but also support the basis for the development of successful periodontitis treatment using exosomes [71]. Further tissue infection could also be avoided by utilizing the antibacterial characteristics of chitosan-based hydrogels [72].

A novel alternative cell therapy technique for periodontal tissue regeneration in periodontitis treatment uses a biological injectable hydrogel. Shi et al. developed a gelatin and laponite injectable hydrogel material loaded with lipopolysaccharide-preconditioned dental follicle cell-derived small extracellular vesicles and applied it for the treatment of periodontitis. They found that the hydrogel helped regenerate lost alveolar bone during the early stages of treatment that was retained through the late stages of treatment, partially by reduction of the RANKL/OPG ratio [73]. Similarly, Liu et al. developed a blended laponite/gelatin injectable hydrogel loaded with bone marrow mesenchymal stem cell-derived small extracellular vesicles (BMSC-EVs) and used it as a potential cell-free strategy for regeneration of periodontal tissue in a periodontitis rat model. After administration of hydrogel for 4 to 8 weeks, the BMSC-EVs-hydrogel group exhibited reduced alveolar bone loss, inflammatory infiltration, and collagen breakdown when compared to the PBS-hydrogel and periodontitis groups. The hydrogel also had a constant release of the BMSC-sEVs at 30 days, which confirmed that the hydrogel supports sustained release. The results revealed that the sustained release of BMSC-EVs from the hydrogel can stimulate periodontal tissue regeneration, which may be partially attributed to their involvement in the OPG-RANKL-RANK signaling pathway. This pathway controls osteoclast function, macrophage polarization, and TGF-β1 expression to modulate the inflammatory immune response and prevent periodontitis-induced damage to periodontal tissue [74].

**Figure 5 ijms-25-11092-f005:**
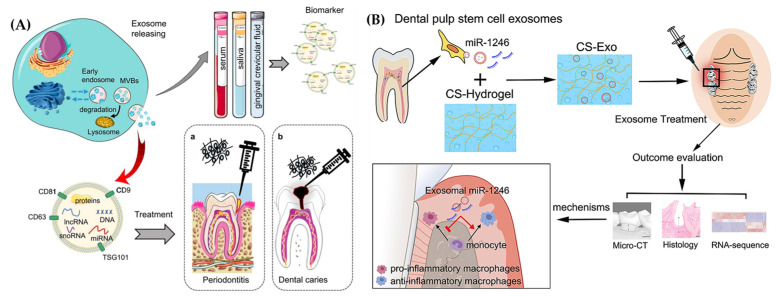
(**A**) Demonstrated use of secreted exosomes for dental tissue regeneration and as biomarkers for periodontitis and dental caries (Reprinted from [56] with permission from Elsevier). (**B**) The development of chitosan hydrogel loaded with exosomes derived from dental pulp stem cells to aid in healing periodontal tissues damaged by periodontitis (Reprinted from [71]).

In a rat model of periodontal abnormalities, collagen sponges loaded with mesenchymal stem cell-derived exosomes (MSC-Exos) have been reported to improve periodontal regeneration without adverse effects. It was found that MSC-Exos within the collagen polymer matrix enhanced the migration and proliferation of periodontal ligament cells by activating prosurvival AKT and ERK signaling through CD73-mediated adenosine receptor activation. This finding demonstrated that MSCs-Exo-loaded collagen promoted periodontal regeneration, potentially through accelerating cell migration and proliferation and provided a basis for the development of a cell-free drug delivery strategy for periodontal defects or injuries [75]. Overall, different types of hydrogels embedded with exosomes offer a viable cell-free therapeutic option for use in oral tissue engineering, particularly for the treatment of periodontitis.

## 3. Bone Tissue Engineering

Bone is a hierarchically organized and porous tissue with layers of cortical and cancellous bones, as illustrated in Figure 6A. At the center, cancellous bone (also known as trabecular or spongy bone) provides a structural scaffold to support the bone marrow encased within it. Both the marrow and cancellous bone are surrounded by a layer of compact cortical bone that is responsible for strength and mechanical stability. The surface of cortical bone is covered with an additional layer of tissue called the periosteum, comprised mainly of osteoblasts and blood vessels that are necessary during growth and repair [76]. The entire adult skeleton is composed of about 80% cortical bone and 20% cancellous bone at varying proportions throughout the body. Due to the essential nature of biological processes that occur within bone, its composition is critical to supporting proper function and maintenance, most of which are dependent on the ratio between organic and mineral components within its matrix. The organic component is comprised of collagen, noncollagenous protein, proteoglycans, sialoproteins, glycoproteins and “gla” proteins that account for approximately 25–30% of the matrix and is complemented by the mineral component, hydroxylapatite, which makes up the remaining matrix. From a materials science perspective, these variations in composition and mechanical properties allow bone to be considered as a composite material [77].

Bone tissue engineering is an effective therapeutic approach for the treatment of acute or progressive bone abnormalities, including traumatic injuries, cancer, and congenital malformations [78]. While the gold standard for bone regeneration is autologous bone grafting, this approach carries the risk of significant complications ranging from pain at the donor site to rejection or disease transfer. Biomaterials and bioactive scaffolds are promising alternatives due to their biocompatibility and mechanical and osteogenic properties that are comparable to native bone [9]. Biomineralization is required for proper integration of biomaterials with native tissue in bone tissue engineering applications. The use of hydrogels with nanofillers (e.g., hydroxyapatite or bioactive glass) that reinforce biomineralization would not only encourage increased osteointegration but also provide a vehicle for various constituents, such as biopolymers, synthetic polymers, stem cells, growth factors, nanoparticles, and bioactive drugs, to increase biological function (Figure 6B) [79,80,81,82]. With an increased number of research groups working to enhance the biological characteristics of biomaterials or create new applications in the form of 3D-printed scaffolds, bone tissue engineering presents a wide range of possible therapeutic approaches for the regeneration of bone [9,83].


**Figure 6 ijms-25-11092-f006:**
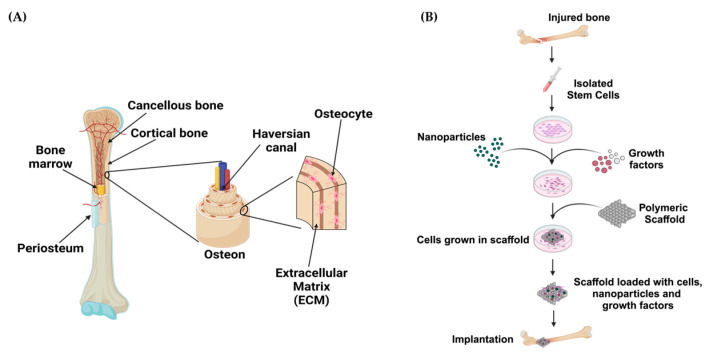
(**A**) Illustration of the bone, showing the cellular distribution and overall structure. Osteoprogenitors are abundant in the bone marrow and periosteum and play important roles in bone repair. In addition, a considerable degree of vascularization is seen in the periosteal and intramedullary canals of the bone (Adapted from [76]). (**B**) Steps for the implantation of biomaterial-loaded scaffolds for bone tissue engineering. Mesenchymal cells are isolated from the donor, cultured in vitro to differentiate into osteoblasts, and then loaded onto scaffolds that contain growth factors, polymers, biomaterials, nanoparticles, etc., prior to implantation. Following their implantation, the scaffolds could promote bone healing and regeneration (Adapted from [81]).

Improved understanding of the embryonic development of bone, fracture healing and its hierarchical structure has facilitated the design of novel bone tissue engineering materials and techniques that mimic natural tissue by both supporting growth and re-storing functionality [84]. The combination of biomaterials and different cell types, including bone marrow mesenchymal stem cells (BMMSCs) and primary adult osteoblasts, has shown great potential for bone regeneration in animal models. However, most of the research focuses on rodents with limited studies on bone defects in larger mammals and limitations in the predictive capacity of the rodent model have hindered the translation to human trials. This area of bone tissue engineering faces not only technical but also ethical challenges in the form of cell sources and safety, thus, the overarching goal for biomaterials is to fill the bone defect and provide a conduit for recipient cells to repair or reconstruct the defect [82,84]. Within bone, the functional cell types contrast the ECM they are encased in, which not only provides mechanical stability but can also have embedded bioactive molecules secreted by cells. In the context of bone regeneration, biomaterials are often fabricated in a scaffold format to temporarily fill the role of the ECM to support cell adhesion and subsequent deposition of a mineralized matrix [84]. The unique properties of 3D structural nanocomposites that promote the natural bone healing process and resemble native structures have gained attention as replacements for autografts and their fabrication techniques generate the potential for more individualized yet functional scaffolds. Despite notable advancements and the ability of bone tissue engineering approaches to mitigate major complications associated with bone grafts, the use of biomaterials in routine clinical practice still faces numerous obstacles related to safety and efficacy that require additional investigation with human clinical trials [85].

### 3.1. Hydrogels for Bone Tissue Engineering

Hydrogels are a suitable biomaterial that is frequently employed in bone tissue engineering due to their porous structure, significant biocompatibility, and similarity to the extracellular matrix [86,87]. Injectable hydrogel-based drug/cell product delivery has become a viable solution in regenerative medicine, where the restoration of bone loss is crucial to the healing process. Hydrogels could prove invaluable in treating arthritis or cartilage injury due to their high absorbance and hydrophilic three-dimensional porous framework (Figure 7) [88,89,90,91]. The similarity of hydrogel scaffolds’ chemical and physical composition to natural bone offers specific advantages for the stimulation of stem cell osteogenic functions [92]. In addition to drug delivery systems, such as oral administration, cutaneous penetration, and/or intravenous injection, numerous composite hydrogels have been developed for implants based on the type or location of bone disease. Among them, implanted hydrogels offer several benefits in clinical applications, such as adjustable shape at the intended location and improved mechanical stability. These characteristics are particularly useful for tissues that are load-bearing sites or have large defects. Due to the outstanding sol–gel characteristics of in situ injectable hydrogel, it can be applied to cartilage defects and minor bone injuries that do not require extensive surgery. The administration of either microgels or nanogels via local injection at the infected region is being investigated as a drug delivery system for the treatment of osteoarthritis (OA), rheumatoid arthritis (RA), and other bone abnormalities [93].

Naturally occurring biopolymers, including chitosan, hyaluronic acid, alginate, collagen, and gelatin, all have the benefit of being nontoxic, biocompatible, and degradable by the human body. Thus, they are frequently employed in bone tissue engineering as functionalized hydrogel materials [69,94]. The structure of chitosan’s polysaccharide unit is comparable to that of glycosaminoglycan, which is the primary building block of cartilage and the extracellular matrix (ECM) in bones. Chitosan (CS) and chitosan-based hydrogels are nontoxic, biocompatible and degradable, properties that are all helpful in the field of biomedicine, particularly bone tissue engineering. Its porous structure also promotes cell attachment and proliferation. A hybrid CS-based hydrogel loaded with bioactive glass nanoparticles was proposed utilizing a freeze-gelation approach. By adjusting the ratio of the nanoparticles, the hybrid hydrogel’s surface area, porosity, and mechanical characteristics can be customized. This could encourage the formation of an apatite layer on the hydrogel’s surface, facilitating direct bone interaction with the implanted hydrogels [95]. Chitosan-embedded particles are often combined with polymeric hydrogels, that could promote bone healing by improving localized cell and drug delivery at the bone defect site [96]. The injectable hydrogels crosslinked by the hydrazone chemistry of hyaluronic acid aldehyde and thiol-Michael reaction have shown better bone regeneration due to their cross-linking chemistry, which forms a reversible covalent bond very quickly (<30 s) [97]. The methacrylated hyaluronic acid–fibrin hydrogel was shown to be beneficial in a three-dimensional environment for improving bone marrow stromal cell (BMSC) proliferation through increased *SOX9* expression [98]. The aldehyde group in HA-based hydrogels can also undergo a Schiff-based reaction with polymers containing amino and hydrazide groups. These hydrogels have been extensively explored in the field of cartilage tissue engineering. The combination of chitosan and hyaluronic acid in hydrogels can provide a favorable environment for chondrocyte proliferation, adhesion, and differentiation. The encapsulated chondrocytes also exhibit a spherical morphology and even distribution throughout the chitosan/hyaluronic acid hydrogel [99].

The application of alginate hydrogels for bone tissue engineering was evaluated due to their high availability, gelation capacity, cost effectiveness, and biocompatibility. Another advantage of alginate materials is that they could provide an adequate niche for cell loading and are injectable due to their inherent ionic crosslinking [100]. The cell-free, void-forming alginate hydrogel demonstrated potential for bone regeneration in a load-bearing, 5-mm critical-sized segmental femoral defect [101]. A new generation of composite collagen hydrogels is currently being used extensively as bone tissue scaffolds via 3D printing, electrospinning, and other methods, as well as an injectable hydrogel scaffold for in situ bone tissue healing and flexible drug delivery systems [102]. In 3D culture, collagen repeatedly been demonstrated to preserve chondrocyte phenotypes. Equally dispersed chondrocytes can also create new cartilage-like tissue by synthesizing ECM [103]. Therefore, collagen and collagen-based biomaterials for bone and cartilage tissue engineering appear to be a natural choice. The collagen-elastin-like polypeptide composite hydrogel can improve osteogenic differentiation in human adipose-derived stem cells (hASCs) and also exhibited advantageous characteristics, such as tensile strength, osteogenic activity, and mineralization [104]. Likewise, collagen mimetic hydrogels improved the chondrogenesis process in human mesenchymal stem cells and ECM synthesis [105]. Gelatin is a denatured and hydrolyzed collagen that can be obtained by treating animal skin and bones (usually porcine skin) with an acidic or basic solution, and then separating the proteins using a thermal process. The benefits of gelatin’s biological activity (RGD sequence) for tissue engineering have been highlighted by the widespread application of gelatin particles as hemostatic agents to repair cartilage and bone deformities. Bone morphogenetic protein-2 (BMP-2) and stromal cell-derived factor-1 (SDF-1) were included in gelatin-based hydrogels and showed that the combined release of growth factors promoted bone regeneration more than their use individually [106].

### 3.2. Exosomes for Bone Tissue Engineering

Bone tissue and cartilage regeneration rely on multicellular activities, including remodeling, differentiation, functional cell recruiting, immunogenicity, and angiogenesis. Exosomes, a key component of cell signaling, are currently a popular topic in the field of bone and cartilage regeneration because of their ability to transport proteins, nucleic acids, and bioactive lipids (Figure 8) [107]. Recent research has demonstrated that exosomes obtained from different cells, such as mesenchymal stromal/stem cells, osteocytes, osteoblasts, and endothelial cells, can be involved in multiple steps of bone regeneration [108,109]. Bone-related miRNAs, extracellular matrix proteins, and some osteogenic proteins have been shown to be transported via exosomes. It is possible to alter parent cells to export specific proteins and miRNAs, enabling their precise delivery to nearby cells [110]. In particular, MSC-Exos have lower concentrations of the cytokines that can promote osteogenesis and other angiogenic factors, such as monocyte chemoattractant protein (MCP)-1, MCP-3, and stromal cell-derived factor (SDF), when compared to conditioned media [111]. Following adenovirus transfection, treatment with exosomes released by BMSCs overexpressing HIF-1α resulted in increased expression of osteogenic markers, such as osteocalcin and alkaline phosphatase [112]. Additionally, treatment with human umbilical cord-derived MSCs-exosomes resulted in improved neovascularization and accelerated bone regeneration in vivo in a steroid-induced model of osteonecrosis of the femoral head in rats and a femoral fracture model in mice and rats [110]. Cartilaginous connective tissue, mostly found in joints, has a limited ability to regenerate on its own after damage or injury. Conditions including rheumatoid arthritis (RA) and osteoarthritis (OA) that affect joints may exacerbate these injuries. Synovial fluid or synovial fibroblast-derived extracellular vesicles (SFB-EVs) are the main candidates under investigation to clarify the mechanisms of joint disease and associated cartilage damage since synovial fluid and integrated SFBs may cause inflammatory propagation supporting degeneration of cartilage [107]. Exosomes produced from MSCs have shown potential in cartilage regeneration, with studies indicating that treating animal models of osteochondral defects or OA with MSC-Exos improved cartilage regeneration through improved extracellular matrix deposition, cellular migration, proliferation, and histological scores [110,113].

### 3.3. Exosome Laden Hydrogels for Bone Tissue Engineering

Hydrogels have a 3D network structure that mimics the natural ECM and a high water absorption capacity, which supports their use for widespread tissue engineering applications. Exosomes aid in a number of physiological signaling cascades, including angiogenesis, wound healing, and cell differentiation. Therefore, recent innovative cell-free tissue engineering methods for bone regeneration that employ exosome-laden hydrogels have been developed and extensively studied [114]. These cell-free methods allow for local, concentrated delivery of the therapeutic compounds contained in exosomes [114,115]. For example, periodontal ligament stem cell-derived exosomes (PDLSC-Exos) loaded on a gelatin-sodium alginate hydrogel promoted more alveolar bone regeneration compared to exosome-free hydrogels [116]. While hyaluronic acid-based hydrogels have recently shown encouraging results in bone regeneration, tissue restoration requires more than just a HA scaffold within the injured area. Consequently, different cell-derived exosomes have been effectively added to hyaluronic acid-based hydrogels to improve bone regeneration at the defect sites [116]. The study by Zhang et al. developed hyaluronic acid-based hydrogels that are loaded with exosomes obtained from umbilical mesenchymal stem cells (uMSC-Exos) and combined with nanohydroxyapatite/poly-ε-caprolactone (nHP) scaffolds for cranial defects in a rat model (Figure 9A). The results indicate that the exosome-laden hydrogel promoted angiogenesis during the healing of large bone defects, potentially through the miR-21/NOTCH1/DLL4 signaling pathway [117].

A dangerous bone condition known as glucocorticoid-induced osteonecrosis of the femoral head (GC-ONFH) frequently affects adolescents. The most common clinical treatment for GC-ONFH is bone grafting in conjunction with core decompression. Chen et al. developed a hydrogel with a combination of methacrylated type I collagen and exosomes isolated from bone marrow stem cells (BMSCs) stimulated with lithium ions (Li) to treat GC-ONFH in a rat model. The study demonstrated that the Li-Exo hydrogel had the most positive effect in improving angiogenesis, osteogenesis, and M2 macrophage polarization to promote bone repair in the GC-ONFH model. This also revealed that this unique hydrogel mimics the ECM and when functionalized with exosomes might serve as a promising therapeutic for osteonecrosis [118]. Another study synthesized a self-healing hydrogel with coralline hydroxyapatite (CHA), silk fibroin (SF), glucose chitosan (GCS), and difunctionalized polyethylene glycol (DF-PEG) as carrier for human umbilical cord mesenchymal stem cells (hucMSC)-derived exosomes carrier to repair bone defects in Sprague-Dawley (SD) rats (Figure 9B). The hydrogel had shown excellent physical and biological properties, such as self-healing, smooth surface morphology, spherical crystal structures, and good biocompatibility with mouse osteoblast progenitor cells (mOPCs). The results of in vivo studies indicate that rat bone defects could be repaired more quickly when the hydrogels containing exosomes are used. The histological investigation also showed more newly formed bone, BMP2 expression, and had the highest microvessel density [119].

**Figure 9 ijms-25-11092-f009:**
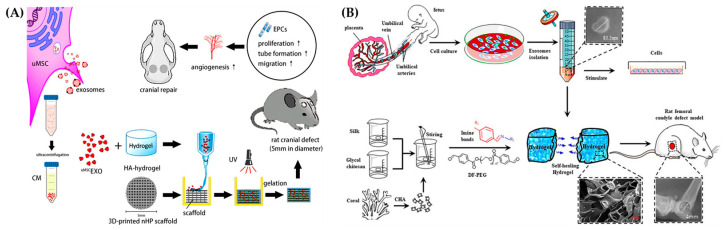
(**A**) Schematic illustration of uMSCEXO isolation for HA-Gel embedding and nHP scaffold printing to heal the critical-size cranial defect in rats by promoting angiogenesis (Reprinted with permission from [117]). (**B**). Schematic illustration of the isolation and characterization of hucMSC-derived exosomes and synthesis of CHA/SF/GCS/DF-PEG hydrogel with exosomes for testing in Sprague–Dawley rats with an induced femoral condyle defect (Reprinted from [119]).

The efficacy of hydrogels to control drug release and improve therapeutic outcomes has been enhanced through the manipulation of their hydrophilic and cross-linking behaviors [120]. Chitosan has received great attention due to its extrinsic biomineralization and antimicrobial properties. Studies on bone tissue engineering have favored the chitosan and β-glycerophosphate (CS-β-GP) hydrogels due to their injectability and thermosensitive characteristics [121]. A study by Wu et al. designed a CS-β-GP injectable and thermosensitive hydrogel with encapsulated exosomes isolated from BMSCs that significantly improved the repair of calvarial defects in a rat model through prolonged delivery and significant biocompatibility. In bone defects implanted with the hydrogel, the expression of osteogenic gene alkaline phosphatase (ALP) was enhanced, mineral deposition was increased, and the area of new bone formation was expanded [122]. Similarly, in vitro and in vivo studies have demonstrated the therapeutic effects of UV-responsive gelatine methacrylate (GelMA) hydrogels embedded with exosomes derived from human periodontal ligament fibroblasts (hPDLFs) for calvarial defects in a rat model (Figure 10A,B). The application of this hydrogel showed an upregulation of *OSP*, *ALP*, and *RUNX2* expression, indicating the stimulation of osteogenic differentiation in hMSCs without the addition of growth factors. The gene expression changes, histochemistry, and micro-computed tomography (μ-CT) results demonstrate that rats treated with GelMA/hPDLF-Exo hydrogel produced more new bone mineralization than the negative controls [123]. Huang et al. also developed photocrosslinkable alginate hydrogels with an RGD peptide and mesenchymal stem cell-derived extracellular vesicles (MSC-EVs) with tissue regenerative capabilities, specifically the stimulation of bone regeneration in a rat calvarial defect model with prolonged delivery in vivo [124].

A highly prevalent cause of disability in the elderly is osteoarthritis (OA), a bone disease that can cause joint pain, stiffness, and reduced function. Degenerative cartilage lesions are the main feature of OA. Highly differentiated cells, chondrocytes, respond to stimuli mainly through the production and secretion of the cartilage matrix, which is essential for maintaining the metabolic balance of cartilaginous tissue. Currently, there are no drugs available to stop the progression of OA. Injecting therapeutic materials, specifically, hydrogels with encapsulated exosomes have been investigated as a new treatment method for osteoarthritis and cartilage regeneration [125]. Liu et al. developed a photoinduced imine crosslinking (PIC) hydrogel with o-nitrobenzyl alcohol moieties and modified hyaluronic acids (HA-NB)-gelatin-human induced pluripotent stem cells (hiPSC)-derived MSC-Exos for cartilage regeneration (Figure 10C). The hydrogel was found to positively regulate both hBMSCs and chondrocytes in vitro and may interact with the natural cartilage matrix to facilitate the accumulation of cells in cartilage defect regions, which aids in the regeneration of cartilage. This material could offer a novel and cell-free approach for cartilage tissue engineering [126]. The sustained release of exosomes from in-situ crosslinking of hyaluronic acid/Pluronic F-127 injectable and a thermosensitive hydrogel loaded with chondrocyte-derived exosomes was found to promote cartilage matrix formation and prevent cartilage destruction. This occurred through positive regulation of chondrocyte proliferation, migration, and differentiation, as well as effective stimulation of M1 to M2 macrophage polarization, offering another potential treatment for osteoarthritis [127]. A study from Zhang et al. reported the possible treatment for cartilage abnormalities in rat patellar grooves using an adhesive and injectable hydrogel fabricated with regenerated silk fibroin, chondroitin sulfate, alginate–dopamine, and BMSC-Exos. After encapsulating the exosomes, the hydrogel promoted migration, proliferation, and differentiation of BMSCs without adverse effects. Notably, the adhesive hydrogel stimulated the differentiation of BMSCs into chondrocytes and promoted the repair of cartilage defects in the rat patellar grooves by recruiting endogenous BMSCs to the defect via chemokine signaling pathways [128]. Overall, experimental outcomes suggested that exosome-embedded hydrogels have potential as biomaterials for bone tissue engineering, including cranial, calvarial, and cartilage regeneration.

**Figure 10 ijms-25-11092-f010:**
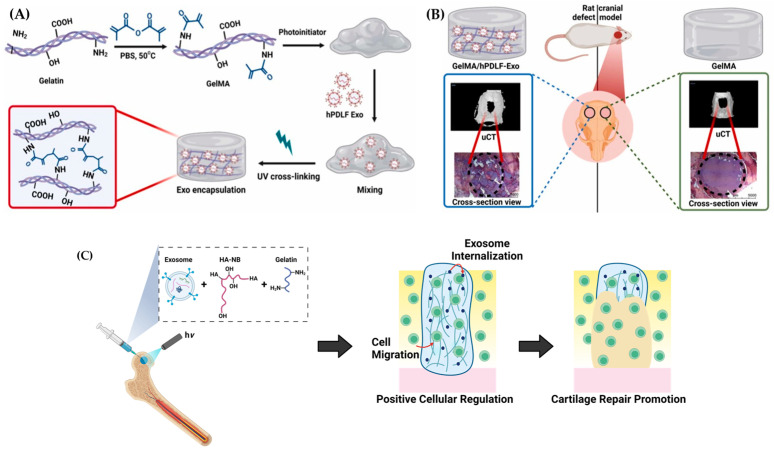
A rationale for the design of exosome-laden hydrogels. (**A**) Fabrication of GelMA with subsequent hPDLF-Exo embedding and hydrogelation via photo-crosslinking. (**B**) Application of GelMA (right) and GelMA/hPDLFs-Exo (left) hydrogels into calvarial defects in a rat model (Reprinted from [123] with permission from Elsevier). (**C**) Schematic illustration of photoinduced imine crosslinking (PIC) hydrogel integrated with human induced pluripotent stem cells (hiPSC) derived exosomes for cartilage regeneration (Adapted from [126]).

## 4. Methods

To write this review article, a scientific literature search was conducted using various databases, such as ScienceDirect, Scopus, Web of Science, Pubmed, and Google Scholar, with the following keywords: biomaterials, hydrogels, exosomes, oral, bone, and cartilage tissue engineering, and regenerative medicine. By utilizing these keyword combinations, we have been able to locate significant previous investigations in the field. The therapeutic efficacy of exosome-laden hydrogels for bone and oral tissue engineering was specifically reported in these publications. The intention is to give the reader a general introduction to this area of research, while also highlighting the advantages and potential behind novel combinations of cutting-edge and conventional biomaterials.

## 5. Conclusions

Oral and bone defects are difficult to repair due to the complex three-dimensional structure required and the potential to cause considerable damage to surrounding structures, resulting in deformities and impaired function. Existing treatments are only able to prevent the progression of disease and enhance clinical diagnostic markers but cannot completely restore the tissue and organ functions. Therefore, new biomedical approaches are required to overcome these challenges to achieve successful craniofacial bone and oral tissue regeneration at the morphological, structural, and functional levels. The development of hydrogel biomaterials, their fabrication techniques, and biomimetic characteristics have been aided by progress in oral and bone tissue engineering and regenerative medicine.

Despite progress with biomaterials, there are several clinical requirements that need to be met for biological and biomedical applications. Thus, the encapsulation of exosomes in hydrogels has offered a novel opportunity to promote the effective healing and repair of oral and bone tissues through biocompatible and sustained cell-free drug delivery systems. The encapsulation of therapeutic agents, such as exosomes, into hydrogels can have various effects on biological processes that are essential for the dental and bone regeneration processes, including angiogenesis, inflammation, immune regulation, and osteogenic differentiation.

In this review, we summarized the therapeutic effects of exosome-laden hydrogels for oral and bone tissue engineering as a cell-free drug delivery system. Cell-free hydrogel therapies, utilizing exosomes isolated from different cell sources, have shown positive effects on the mineralization process for both craniofacial bone and oral tissue regeneration, projected to have an added cost-effective benefit.

In conclusion, the application of modified biomaterials, namely exosome-laden hydrogels, is a burgeoning area of research, and this new therapeutic approach will be beneficial to creating viable, long-term tissue repair methods for patients.

## Figures and Tables

**Figure 1 ijms-25-11092-f001:**
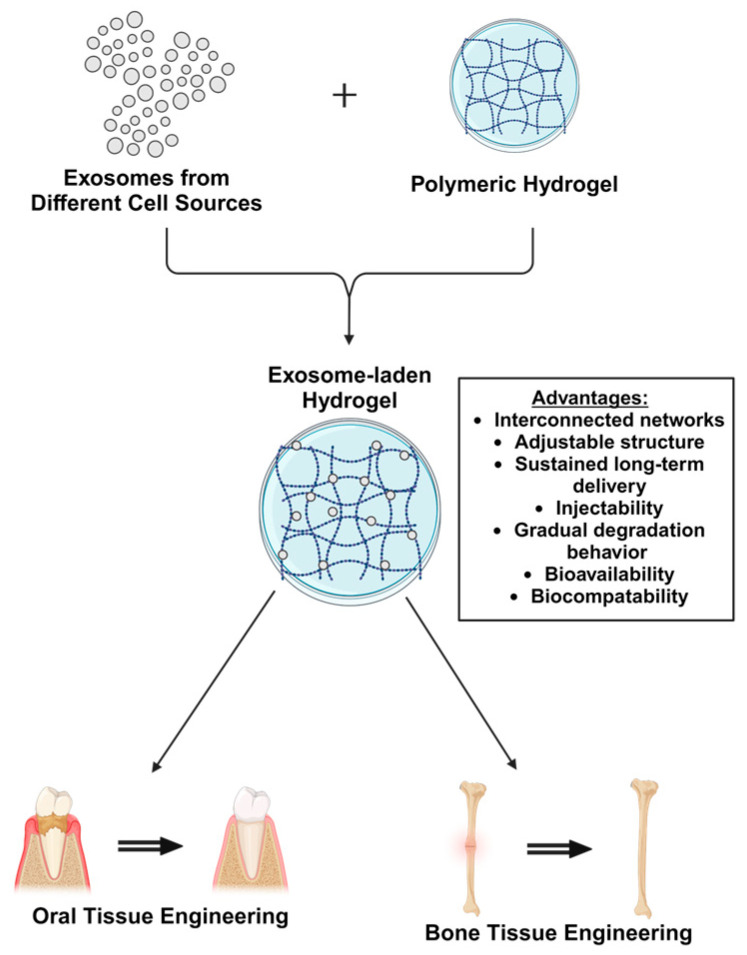
Schematic illustration of exosome-laden hydrogels for bone and oral tissue engineering.

**Figure 2 ijms-25-11092-f002:**
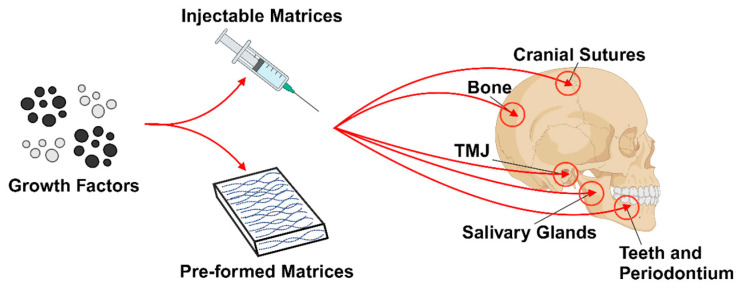
Drug delivery in tissue engineering applications for dental, oral, and craniofacial regeneration. In situ formation of injectable matrices or preshaped implants can be used to provide growth factors or other bioactive molecules for the regeneration of teeth, periodontal tissues, temporomandibular joints, cranial sutures, salivary glands and calvarial bone (Adapted from [31]).

**Figure 3 ijms-25-11092-f003:**
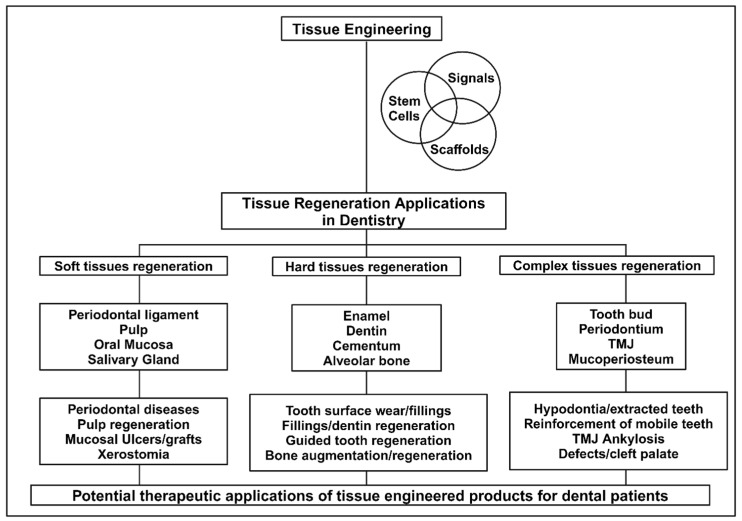
Clinical and therapeutic applications of oral tissue engineering in dentistry (Adapted from [28]).

**Figure 4 ijms-25-11092-f004:**
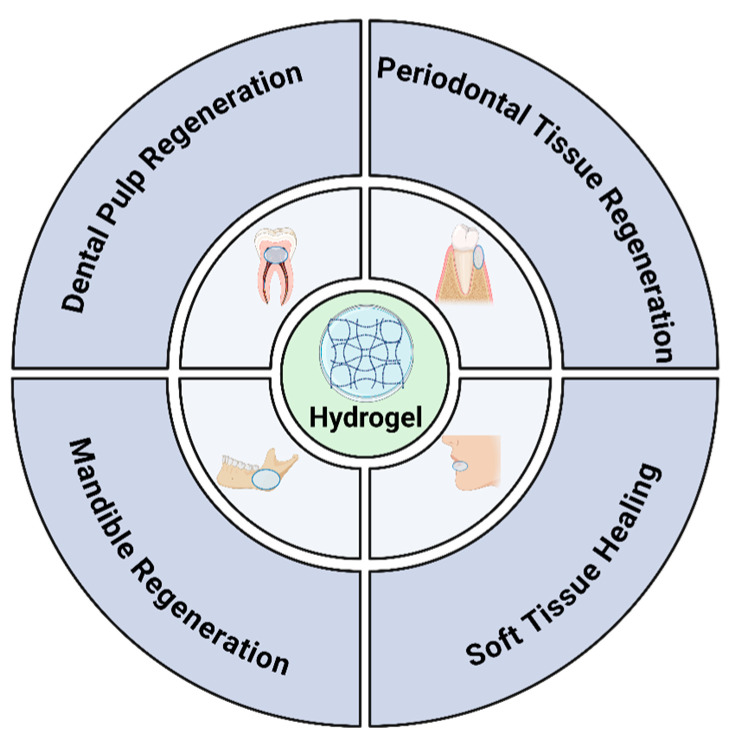
Applications of engineered hydrogels for oral tissue regeneration (Adapted from [42]).

**Figure 7 ijms-25-11092-f007:**
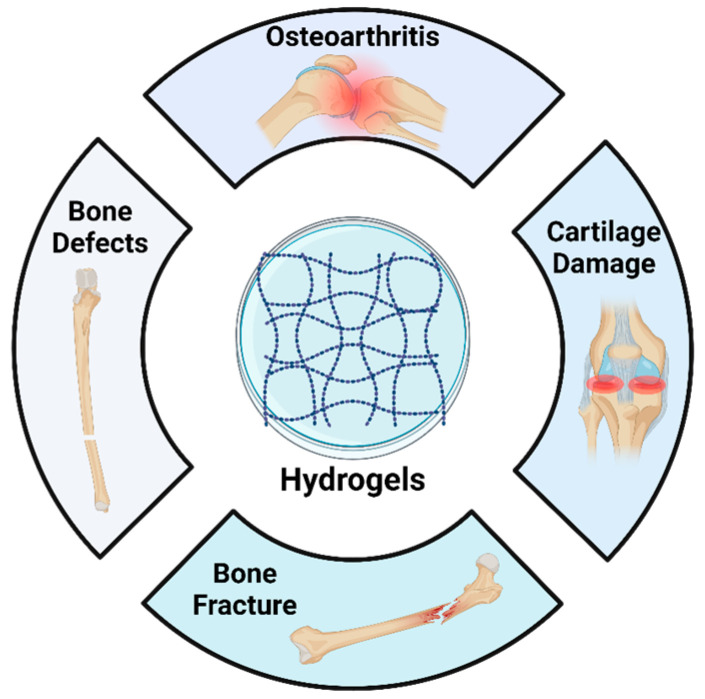
Applications of hydrogels in bone tissue engineering (Adapted from [88]).

**Figure 8 ijms-25-11092-f008:**
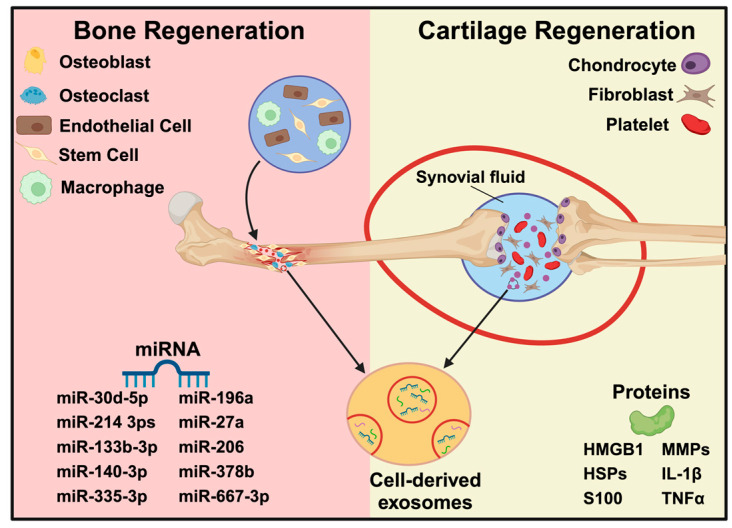
Illustration of cell-derived exosomes with cargo during bone and cartilage regeneration (Adapted from [107]).

## Data Availability

Not applicable.

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
