# Peer review of "Exosome-Laden Hydrogels as Promising Carriers for Oral and Bone Tissue Engineering: Insight into Cell-Free Drug Delivery"

_ijms, 2024, doi:10.3390/ijms252011092_

Round 1
Reviewer 1 Report
Comments and Suggestions for Authors
The review by Villani et al. reports on the most recent advances in the application of hydrogels loaded with exosomes and extracellular vesicles for oral and bone tissue engineerin. Hydrogels of different compositions have been employed in tissue engineering, but numerous evidences have shown that better results can be obtained when vesicles of cellular origin are included in the hydrogel structure, due to the properties of cross-talk that these vesicles have proved to possess.
In my opinion this review is very well organized. Maybe in several sections the introductive part is a little bit too long, but I understand it can be useful to introduce the topic to scientists not working in the field. The cited references are sufficient and all pertinent to the subject. I would only suggest the Authors to add a few lines regarding the drawbacks of applying exosome-laden hydrogels to tissue engineering (reproducibility, costs, tolerability, stability, etc.).
A few observations:
line 44 maybe the verb is missing
line 69 "periods" to be replaced with "period"
line 106 "and not only controlled but sustained release at the target site": actually "sustained release" is normally referred to a prolonged, slow release, while "controlled release" refers to systems that are actively able to generate a constant release rate, therefore I would suggest to change this sentence into "and not only sustained but also controlled release at the target site"
line 154 I think that reminding that permanent dentition is not replaced is redundant
lines 291 and 314 hDPSC has been already defined as acronym
line 445 "requires" to be replaced with "require"
line 457 "of" must be removed
line 546 "SF" has been used for "sinovial fluid" and for "silk fibroin": this is confusing
line 567 maybe a "compared to" is missing
line 620 the hydrogel is not loaded with fibroblasts, but rather with fibroblast exosomes: this must be amended
Comments on the Quality of English LanguageHigh quality English, just a few typos to be corrected
Author Response
Comment 1: I would only suggest the Authors to add a few lines regarding the drawbacks of applying exosome-laden hydrogels to tissue engineering (reproducibility, costs, tolerability, stability, etc.).
Response 1: Thank you, this has been included at line 106
Comment 2: line 44 maybe the verb is missing.
Response 2: Thank you, the sentence has been rephrased for clarity.
Comment 3: line 69 "periods" to be replaced with "period".
Response 3: Thank you, this has been corrected.
Comment 4: line 106 "and not only controlled but sustained release at the target site": actually "sustained release" is normally referred to a prolonged, slow release, while "controlled release" refers to systems that are actively able to generate a constant release rate, therefore I would suggest to change this sentence into "and not only sustained but also controlled release at the target site"
Response 4: Thank you, the suggested adjustment has been made.
Comment 5: line 154 I think that reminding that permanent dentition is not replaced is redundant.
Response 5: Thank you for pointing this out, we agree the original sentence did seem redundant and has been rephrased
Comment 6: lines 291 and 314 hDPSC has been already defined as acronym
Response 6: Thank you, the repetitive definitions have been removed.
Comment 7: line 445 "requires" to be replaced with "require".
Response 7: In line 445 ‘requires’ is talking about the use of more than one biomaterial and thus needs to remain plural.
Comment 8: line 457 "of" must be removed
Response 8: Thank you, this has been removed.
Comment 9: line 546 "SF" has been used for "sinovial fluid" and for "silk fibroin":
Response 9: Thank you, the repetitive definitions have been removed and adjusted accordingly
Comment 10: line 567 maybe a "compared to" is missing.
Response 10: Yes, this has been added
Comment 11: line 620 the hydrogel is not loaded with fibroblasts, but rather with fibroblast exosomes: this must be amended.
Response 11: Thank you, this has been amended appropriately.
Reviewer 2 Report
Comments and Suggestions for Authors
Considering the application of hydrogel in regenerative medicine and the reliability of exosomes in repair. This article aims to study the prospects of combining the two, so it is recommended for publication. However, there are still some issues in this article that need to be improved and rectified.
Comments:
1. It is suggested to add relevant literature and treatment schemes of combined exosomes and hydrogel therapy in the field of regenerative medicine in the introduction, rather than merely stating the defection of exosomes and the advantages of combined exosomes and hydrogel therapy.
2. In line 174, examples and methods of 3D printed scaffolds in the medical field for repairing tissue defects in the oral cavity can be added in detail.
3. From line 188 to line 189, it is recommended to adapt the clinical and therapeutic applications of oral tissue engineering in dentistry.
4. From line 208 to line 209, the author proposes that hydrogel can promote wound healing by improving the magnetic field in the surrounding environment of the tissue. Please explain its principle and supporting references in detail.
5. Please provide a detailed explanation on lines 276, 277, and 278 regarding the promotion of periodontitis by DSPE Exos, the in vivo repair mechanism of rat periodontal epi-278 epithelium and alveolar bone, as well as relevant references and supporting evidence.
6. The formation, extraction methods, biomarkers, and structural images of extracellular vesicles in line 358 are not clear. It is recommended to improve the resolution of the images and provide detailed textual explanations.
Comments on the Quality of English Language
N/A
Author Response
Comment 1: It is suggested to add relevant literature and treatment schemes of combined exosome and hydrogel therapy in the field of regenerative medicine in the introduction, rather than merely stating the defection of exosomes and the advantages of combined exosomes and hydrogel therapy.
Response 1: Thank you, information on combined exosome and hydrogel therapy along with corresponding literature has been adding to the introductory paragraph beginning with line 113.
Comment 2: In line 174, examples and methods of 3D printed scaffolds in the medical field for repairing tissue defects in the oral cavity can be added in detail.
Response 2: Thank you, additional detail and citations on 3D printed scaffolds has been added starting at line 193
Comment 3: From line 188 to line 189, it is recommended to adapt the clinical and therapeutic applications of oral tissue engineering in dentistry.
Response 3: Lines 188-189 in the original draft referred to the figure legend for figure 3 which highlights tissue engineering adaptations specifically for dentistry.
Comment 4: From line 208 to line 209, the author proposes that hydrogel can promote wound healing by improving the magnetic field in the surrounding environment of the tissue. Please explain its principle and supporting references in detail.
Response 4: Thank you for pointing this out - Line 209 states that the modulation of magnetic fields can be used to better control release of molecules from a hydrogel which would then aid in wound healing, this doesn’t mean that the magnetic field alterations would directly results in better wound healing. We have including additional clarification at line 228.
Comment 5: Please provide a detailed explanation on lines 276, 277, and 278 regarding the promotion of periodontitis by DSPE Exos, the in vivo repair mechanism of rat periodontal epi-278 epithelium and alveolar bone, as well as relevant references and supporting evidence.
Response 5: The exosomes mentioned did not promote periodontitis but rather the improvement on the disease phenotype. This has been further clarified in text starting at line 303.
Comment 6: The formation, extraction methods, biomarkers, and structural images of extracellular vesicles in line 358 are not clear. It is recommended to improve the resolution of the images and provide detailed textual explanations.
Response 6: Thank you, the resolution for images in Figure 5 has been increased; they depict an example of a study mentioned in the article but the technical specifications of extraction or biomarkers used isn’t relevant to expand but was used to highlight a workflow of exosome application.